# RPL Neutron Dosimetry in n-γ Fields in Comparison with Polymer Detectors Type CR-39

**DOI:** 10.3390/polym14091801

**Published:** 2022-04-28

**Authors:** Youbba Ould Salem, Halima Elazhar, Issiaka Traore, Jonathan Riffaud, Abdelmjid Nourreddine

**Affiliations:** 1Institut Pluridisciplinaire Hubert Curien, UMR 7178 CNRS, Université de Strasbourg, CEDEX 2, F-67037 Strasbourg, France; yo.salem@unistra.fr (Y.O.S.); elazhar.halima@gmail.com (H.E.); riffaud@unistra.fr (J.R.); 2Département de Physique, Université des Sciences, des Techniques et Technologies de Bamako, Bamako BP E3206, Mali; traoreamenophis@gmail.com

**Keywords:** RPL detector, CR-39 detector, neutron dosimetry, chemical etching, polymer

## Abstract

Previously we characterized radiophotoluminescent (RPL) detectors for measuring both fast and thermal neutrons for personal monitoring. The linear response and angular dependence, which satisfies the ISO 21909 standard makes their application possible in neutron dosimetry. The polymer CR-39 track detectors remain one of the most used dosimeters where neutron radiations are to be measured, but the visualization process is time consuming. The difference between results obtained by RPL and CR-39 has been discussed mainly for fast neutrons. The present study has also looked at thermal neutron dosimetry, where we found few results in the literature for CR-39. Our measurements demonstrate that RPL detectors can be advantageously incorporated into a dosimeter to measure thermal neutrons.

## 1. Introduction

Neutron dose monitoring in a mixed n-γ field constitutes a current problem of radiation protection. Today’s instruments do not respond correctly to one or the other of these radiations. The new values of the radiation weighting factor W_R_ of ICRP 103 for neutrons [1] make it useful to set up a new generation of dosimeters. Compared to other methods of radiation detection, the systems based on RPL techniques are part of a category of dosimeters that have passed all the conformity tests [2], and their use has changed the approach to personal dosimetry in France. However, RPL glass has a relatively low sensitivity to neutrons [3], and that is the reason for using photon dosimetry for the γ part of a mixed n-γ field [4]. Aware of the need for neutron monitoring in the nuclear power industry, research, or medical sector, the French Radiation Protection and Nuclear Safety Institute (IRSN) has introduced the RPL neutron dosimeter. It is a combination of an RPL dosimeter (for γ, X, and β radiations) and a CR-39 dosimeter (for neutrons) that furnish an accurate measurement of the dose received by workers in a mixed n-γ field.

CR-39 is a polyallyldiglycol carbonate polymer with chemical properties, which make it one of the most sensitive solid nuclear track detectors in terms of linear energy transfer. Due to its transparency to photon radiations of all energies, it offers many advantages for neutron dosimetry in a mixed n-γ field.

Nuclear track detectors have found many applications and particularly so for the development of neutron dosimeters [5,6,7,8,9,10]. However, the chemical etching processes of the detectors and track counting systems vary according to authors [11,12,13,14,15] and, in any case, long. Hence, it would appear useful to set up standard chemical etching parameters for neutron track detectors or else to develop another system for neutron detection. RPL detectors were recently characterized for measuring neutrons [16,17,18], and it has been suggested that an RPL dosimetry system offers an attractive alternative to neutron dosimeters. The objective of our study is to compare the characteristics of these two neutron detectors (i.e., RPL and CR-39) in regards to their reproducibility, dose response, angular dependence, and energy dependence.

## 2. Etching Nuclear Tracks

Historically, in the late 1950s, it was discovered that when mica is exposed to energetic charged particles, these leave latent traces in the material at the atomic scale. It soon became apparent that glass, polymers, or certain other materials can be treated in the same way. Solid-State Nuclear Tracks Detectors (SSNTD) record their passage when their properties are compatible with detection energy thresholds [19] and they are detected via the observation of the nuclear traces created. These damage traces can be revealed and made visible in an optical microscope by treatment with a properly chosen chemical reagent that rapidly and preferentially attacks the damaged material. It removes the surrounding undamaged matrix less rapidly so as to enlarge the etched holes that mark and characterize the sites of the original, individual damaged regions. Fleisher et al. [20], illustrated the envisioned model of track formation in polymeric solids and sketches how etching proceeds for different tracks. The shapes of the etched tracks result from the simultaneous etching of undamaged material at a general rate v_G_ and the accelerated rate v_T_ along the track. The pits that result have a V-shaped cross-section as sketched, and, therefore, in three dimensions are conical with an angle θ = sin^−1^ (v_G_/v_T_).Figure 1 shows an illustration of a typical process related to each bit generation in the case of neutron detection [21]. In a previous work [22], polymer SSNTD type CR-39 was irradiated with both ^241^Am alpha particles and protons in the energy range of 1.0–5.5 MeV and 0.8–2.8 MeV, respectively. The proton irradiation was made by two different techniques: Rutherford Backscattering and Nuclear Reaction. The track diameter variation as a function of particle energies and etching time has been studied. A predictive model based on experimental track parameters has been developed for the studied particles and for variable bulk track etching velocities v_T_. Figure 1 (right) shows the proton tracks observed by an optical microscope after chemical etching. The counting of the number of traces per unit area to provides the number of incident particles.

## 3. Materials and Methods

### 3.1. RPL Glass Dosimeter

The RPL dosimeter consists of a plastic holder equipped with a flat Ag-doped phosphate glass (35 × 7 × 1.5 mm^3^) associated with filters of aluminium, copper, tin, and plastic that help discriminate the different types of radiations [23]. Information on filter thicknesses was described in detail in a paper published by Huang et al. [24]. RPL glass contains by weight: 48.33% O, 13.24% Na, 6.18% Al, 31.53% P, and 0.72% Ag with a 2.6 g/cm^3^ density [25]. During irradiation, luminescence centers in the glass are activated. Before reading a detector, the centers must undergo stabilizing heat treatment (“preheating’’) at 100 °C for 1 h to accelerate the build-up of colour centers [26]. When illuminated under the UV laser beam of the FGD-660 readout module, the deexcitation of the centers is accompanied by an orange luminescence with an intensity proportional to the number of trapped electrons and thus proportional to the dose. RPL centers do not disappear after a readout operation [27] which allows rereading a detector. Upon heating to 400 °C for 1 h, a detector is erased and can be reused. The highly sensitive reading technique of the reader based on pulsed UV laser excitation enables measuring γ, X, and β radiation from low to high levels of about 10 Gy.

### 3.2. CR-39 Dosimeter

The CR-39 detector mainly used in this work is a Poly-Allyl-Diglycol-Carbonate (PADC) from Technol Chiyoda (Japan) with dimensions of 19 × 8.5 × 0.8 mm^3^ with a 1.29 g/cm^3^ density. Named TechnoTrak2, it is placed in a case made of 1 mm-thick plastic that includes two screens: one is made of polyethylene (for fast neutron detection), and the other is made of a mixture of polyethylene and boron (for thermal neutron detection). During irradiation, charged particles passing through a track detector material induce damage along their trajectory, called latent tracks. A chemical etching process is then used to reveal these latent tracks that can be seen with a microscope [28]. In our case, latent tracks were revealed by chemical etching in a 30% KOH solution at 90 ± 3 °C for 2.5 h. After etching, the detectors were immersed in an HCl solution for 30 min and rinsed in distilled water for 30 min. According to Mukhtar and Qureshi [29], the etchant must be stirred continuously to ensure its homogeneity. Track counting was performed using an automatic analyzing system consisting of a CCD camera mounted on an optical microscope (10× magnification) with a motorized XYZ stage piloted by a PC. RPL neutron dosimeter. As an application, passive dosimetry systems based on CR-39 polymers have recently been used to evaluate activation in food products irradiated with high-energy X-rays [30]. We focused on the detection of neutrons induced by photonuclear reactions likely to occur by X-rays of 7 MeV maximum energy. We were interested in a simple case of irradiation easily reproducible by Monte Carlo simulations. For this, we irradiated a water phantom to generate neutrons and thermalize them. In order to map the neutron flux, various CR-39 detectors were placed at different locations from the incident photon beam. From these measurements, one can conclude that the ratio of thermal to rapid components can vary from 1.24 to 4.64, depending on the position of the detectors. The results obtained with the various dosimeters thus show that neutrons were produced in sufficient quantity to allow validation of the Monte Carlo simulation with this kind of experiment. These results also highlight that the spatial distribution of photoneutrons is quite heterogeneous around and inside the water phantom. This must be taken into account in the activation calculations, in particular when the irradiated samples are themselves also inhomogenous. The experiments carried out thus illustrate that the experimental method using polymer type CR-39 can be used to validate the Monte Carlo modeling of the X-ray spectrum and the Monte Carlo calculation of the spatial distributions of photoneutrons.

### 3.3. RPL Neutron Dosimeter

In previous work [31], we have characterized a passive dosimeter incorporating RPL detectors capable of measuring both fast and thermal neutrons for personal dosimetry. These neutrons can be detected in a mixed n-γ field with appropriate converters. Monte Carlo simulations with MCNPX helped with the geometrical conception of the dosimeter and the choice of materials. The responses of the RPL and their angular dependencies were studied.

#### 3.3.1. Fast Neutrons

Fast neutrons are measured by an assembly of two RPL detectors, of which one is preceded by a 1 mm-thick (CH_2_)_n_ converter. This RPL detector registers ambient γ-rays plus protons resulting from (n, *p*) collisions in the converter. Both RPL detectors are assumed to receive the same quantity of γ radiation. The fast neutron dose measured by the recoiling protons is taken to be the difference between the two RPL readings.

#### 3.3.2. Thermal Neutrons

The (n, α) reaction generally used to measure thermal neutrons is not transposable to RPL detectors in a mixed n-γ field because of the activation of Ag contained in the RPL glass. Thus we use the (n, γ) reaction in a Cd converter. In our case, we have employed two RPL detectors, each one associated with a different thermal neutron converter. The RPL detector preceded by Cd detects γ-rays from the ^113^Cd(n, γ)^114^Cd reaction plus γ-rays of the mixed field and possible charged particles from the other isotopes of Cd. The other RPL detector, preceded by an Al screen and ^10^B converter, registers only γ-rays of the mixed field. The Al stops the products from ^10^B(n, α)^7^Li reactions. The two RPL detector readings furnish sufficient information to quantify both the neutron and ambient γ fields. Knowledge of ambient γ-rays enables finding the number of ^113^Cd(n, γ)^114^Cd γ-rays detected. The whole procedure to quantify the dosimeter response to thermal neutrons is described in our previous paper [18].

### 3.4. Irradiations

Irradiations were performed with the ^241^Am-Be sources of the Institut Pluridisciplinaire Hubert Curien de Strasbourg (IPHC) and the Institut de Physique Nucléaire d’Orsay (IPNO) with activities of 1 and 0.3 Ci, respectively. The ambient neutron dose equivalent rate H*(10) of the IPHC source is 48.8 µSv·h^−1^ at 75 cm, while for the IPNO, this dose equivalent rate is estimated to have the value of 43.74 µSv·h^−1^ at a distance of 45.5 cm. These sources are characterized by their component of fast neutrons. The thermal neutron components are obtained by placing the same sources at the center of a thermalizing nine-inch Bonner Sphere (BS). Cadmium added to the sphere enables absorbing low energies between 0.025 and 0.5 eV. Also, the (^252^Cf + D_2_O) source of the IRSN facility at Cadarache has been used for thermal neutron irradiations. Neutron spectra emitted for each configuration are shown in Figure 2 [32,33]. The (^241^Am-Be + BS) and (^241^Am-Be + BS + Cd) configurations were simulated with the MCNPX code [34].

## 4. Results and Discussion

### 4.1. Reproducibility

The reproducibility for personal regulatory monitoring is important when the RPL and CR-39 detectors are used to estimate the absorbed dose. For each measurement, three RPL detectors were therefore irradiated on a 30 × 30 × 15 cm^3^ ISO water phantom [35] with the IPHC ^241^Am-Be source and three CR-39 detectors with the IPNO ^241^Am-Be source. The reproducibility of the measurements was estimated from the coefficient of variation (CV), defined as the standard deviation divided by the average of the detector response per unit area. The CVs of recoiled protons generated by fast neutrons from ^241^Am-Be for both detectors are shown in Table 1. The reproducibility of CR-39 was better than that of the RPL for fast neutrons. The order of magnitude of these values leads us to assume that the reproducibility of the measured values is not affected when the dose changes from 1 to 10 mSv.

For thermal neutrons, the conversion products are of a different nature. γ-rays from the ^113^Cd(n, γ)^114^Cd and α-particles from ^10^B(n, α)^7^Li were detected with RPL and CR-39 detectors, respectively. The CV values of RPL and CR-39 were 5.8% and 2%, respectively, for a ^241^Am-Be thermalized neutrons dose of 3 mSv. Hsu et al. [36] found a CV below 1% for RPL detectors after irradiation with γ-rays in doses from 0.6 to 20 mGy. Their study takes into account only the reproducibility of readings, while our study uses the measurement of five detectors and n-γ discrimination. Furthermore, as already mentioned, there is the possibility of charged particles from isotopes other than ^113^Cd.

### 4.2. Dose Response

#### 4.2.1. Dosimeter Response in Terms of Ambient Dose Equivalent H*(10)

Figure 3 shows the linearity of RPL and CR-39 detectors to an ambient dose equivalent to H*(10) in the range of 1.3 to 22 mSv of the IPHC ^241^Am-Be neutron source. The dose value on the ordinate was obtained from the calibration factor of each type of detector and their readout at the corresponding dose. The direct proportion is observed between the irradiation dose and readout of detectors.

#### 4.2.2. Dosimeter Response in Terms of Personal Dose Equivalent H_p_(10)

For individual dosimetry, the dosimeters are to be worn on the human body. The calibration procedure must take this into account. So the detectors were irradiated on a 30 × 30 × 15 cm^3^ ISO water-filled phantom [35] to take the albedo effects into account.

*Fast neutrons*: The responses for both RPL and CR-39 detectors with dose are shown in Figure 4. RPL detectors were irradiated in the dose range from 2.2 to 22 mSv of IPHC ^241^Am-Be neutron source and the CR-39 in the dose range from 0.1 to 10 mSv with the IPNO ^241^Am-Be source. It can be mentioned that a simple readout of RPL or CR-39 allows finding the fast neutron exposure dose.

For the RPL detector, the elemental analysis made by Scanning Electron Microscopy showed that the (n, *p*) protons must have an energy greater than or equal to 115 keV to be detected [25]. The absence of the dopant in silver in the first 0–1.7 µm of the front face of the RPL detector will consequently decrease the energy dose distribution. The ideal would be to carry out a surface doping of the RPL glass in order to detect the very low-energy protons.

*Thermal neutrons*: Investigations show that few papers have discussed thermal neutron dosimetry using CR-39 detectors. In the present work, we note the behaviour of these detectors to thermal neutrons in comparison with RPL detectors. Figure 5 shows the RPL detector response to thermal neutrons corresponding to thermal neutron-induced γ-rays from the Cd converter and the CR-39 detector response due to the (n, α) reaction as a function of H_p_(10). RPL detectors were irradiated with ^241^Am-Be thermalized neutrons. For these measurements on CR-39, we have used type PN3 detectors from Landauer with dimensions 20 × 25 × 1.5 mm^3^ with a 1.32 g/cm^3^ density coupled with a natural boron BN1 converter enriched in ^10^B (^10^B 99%). We have irradiated them with (^252^Cf + D_2_O) and (^252^Cf + D_2_O + Cd) sources of the IRSN facility in the dose range from 1 to 4 mSv. The detectors were chemically etched in 6.25 N NaOH at 70 ± 1 °C for 7 h and immersed in dilute HCl for 15 min, and then dried. The difference in track densities per cm^2^ due to (^252^Cf + D_2_O) and (^252^Cf + D_2_O)/Cd sources gives the response of PN3 detectors in terms of α-particles to thermal neutrons [37].

The RPL dosimeter is mainly used for the detection of γ, X, and β radiation. The minimum measurable dose value from ^113^Cd(n, α)^114^Cd γ-rays with the dosimetric system is estimated as being approximately 0.12 mSv [18].

### 4.3. Angular Dependence

The critical effects of SSNTD angular dependencies for passive dosimetry applications have been extensively studied both experimentally and theoretically [38,39]. The angular dependence of both detector materials’ response to neutrons up to 60° to the normal was measured. The responses were obtained by rotating the detector-phantom ensemble around a vertical axis perpendicular to the direction of the incident radiation. The angular dependence results are shown in Figure 6. The responses are normalized to the value of each detector at normal incidence. We can see (Figure 6a) that the CR-39 detector presents a significant angular dependence, as does the RPL detector. For thermal neutrons (Figure 6b), we have an almost identical profile for the RPL detector. However, a critical angular dependence effect for Nuclear Track Detectors is known. Tam et al. [40] indicate that the CR-39 detector angular dependence to fast neutrons from ^252^Cf increases when the source-to-detector distance decreases. In their experimental study of the efficiency of CR-39 to fast neutrons, Makovicka et al. [41] found that the sensitivity decreases when the incidence angle increases. Nevertheless, the arithmetical means of the diminishing RPL detector response for fast and thermal neutrons are, respectively, 6% and 16% lower than the value at normal incidence. These differences satisfy the ISO 21909 [42] requirement stipulating that this difference should not be greater than 30%. Experimentally, for CR-39 detectors, this difference varies, as Makovicka et al. [41] have shown.

### 4.4. Energy Dependence

Neutron dosimeters are conveniently calibrated in reference fields of ^241^Am-Be and ^252^Cf, and the results are used to evaluate the dose from other neutron beams. It is important to know the energy dependence of the dosimeter response. For neutron energies from 0.144 to 19 MeV [32], the response at normal incidence was computed. Figure 7 shows simulated relative responses of both detectors corresponding to the (n, *p*) recoil proton fluence per unit dose. The relative responses are normalized to the responses with the ^241^Am-Be neutron field (average energy fluence 4.16 MeV). We observed that the CR-39 relative response to ^241^Am-Be varies linearly with the energy up to 19 MeV. This is explained by the fact that the bare CR-39 itself is a fast neutron converter. El-Sersy [43] showed that the number of induced recoils in CR-39 detectors depends on the neutron energy. On the other hand, the RPL response shows a slight dependence on neutron energy beyond 4.16 MeV. Therefore, the difference with the CR-39 in energy dependence in the detector responses beyond this energy might be attributed to their different compositions.

For more details, we recall that neutrons are indirect ionizing radiation, and their detection is conducted via converters. In our case, fast neutrons were detected via (n, *p*) reactions in a polyethylene converter and thermal neutrons by (n, γ) reactions in a Cd converter. Photons (indirect ionizing radiation) and protons (direct ionizing radiation) producing the directional dependencies of the RPL detector are known in the literature. The RPL emitted radio-photoluminescent intensity due to the activated luminescence centers in the glass during irradiation is proportional to the amount of radiation received. The probability of having interactions over the entire depth of the RPL detector becomes significant for a normal incident. Indeed, the luminescence centers are created only in a surface layer of the exposed face of the RPL glass or over its entire depth. The absence of the dopant in silver in the first µm of the front face of the RPL detector consequently decreases the RPL intensity when the angle of incidence increases. Moreover, the results for angular dependence correspond to a context of regulatory monitoring. For individual dosimetry, the dosimeter is to be worn on the human body. The calibration procedure must take this into account. So, the dosimeter was irradiated on ISO water-filled phantom (ICRU, 1992, report 47) in terms of the personal dose equivalent. Results were obtained by rotating the dosimeter-phantom ensemble around a vertical axis perpendicular to the direction of the incident radiation.

## 5. Conclusions

This early investigation comparing RPL and CR-39 detectors to measure neutrons shows that the CR-39 has better reproducibility and sensitivity than the RPL but presents an angular dependence that is much more important. The characteristic measurements on RPL detectors make it another choice for measuring neutron dose. In comparison to CR-39, the advantages of the RPL system for neutron dosimetry are rapid exploitation of measurements, reusability of detectors, and low energy dependence in high-energy neutron fields. Consequently, an RPL dosimetry system offers an alternative to other dosimeters, such as those employing, among others, nuclear emulsions, track detectors, superheated emulsions, and photostimulable imaging plates, or thermoluminescence. It would be helpful to investigate a methodology to design a new RPL glass badge to include neutron radiations.

Nevertheless, the experimental response of CR-39 in terms of dose equivalent is linear in the presence of a neutron spectrum with a fast and thermal component. As a result, it can be advantageously used to measure neutron doses in an n-γ field with a high flux of photons.

## Figures and Tables

**Figure 1 polymers-14-01801-f001:**
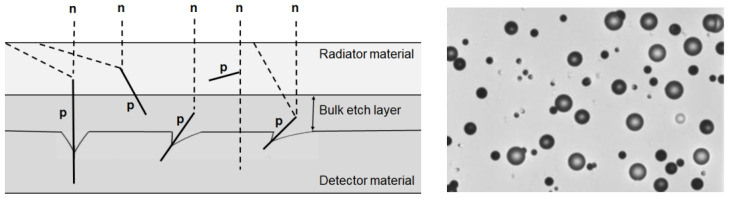
Cross-sectional view with radiator material irradiated by incident neutrons (**left**) and image of pits due to recul protons in a CR-39 track detector (**right**).

**Figure 2 polymers-14-01801-f002:**
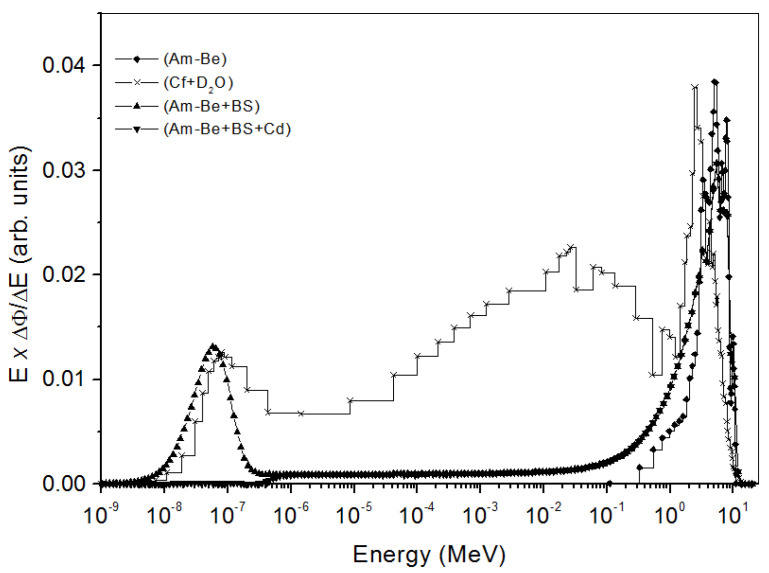
Neutron spectra emitted from ^241^Am−Be, ^241^Am−Be + BS, (^241^Am−Be + BS + Cd), and (^252^Cf + D_2_O) sources. BS is Bonner Sphere.

**Figure 3 polymers-14-01801-f003:**
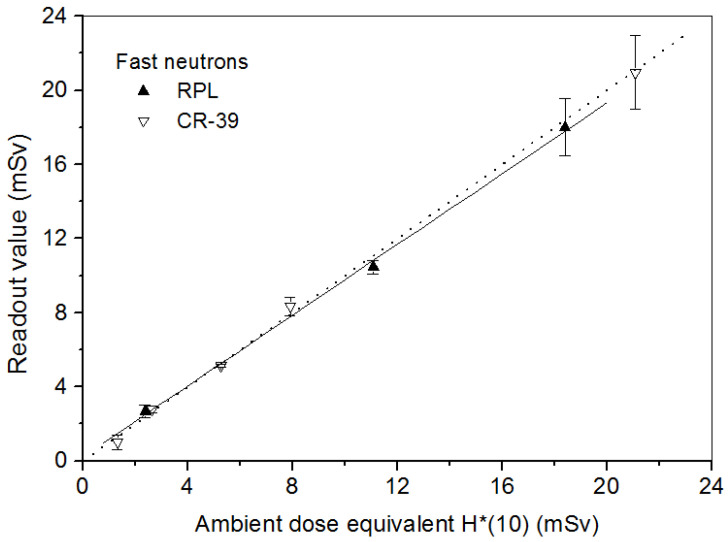
Linear responses of the RPL and CR-39 detectors to ^241^Am-Be neutrons at normal incidence.

**Figure 4 polymers-14-01801-f004:**
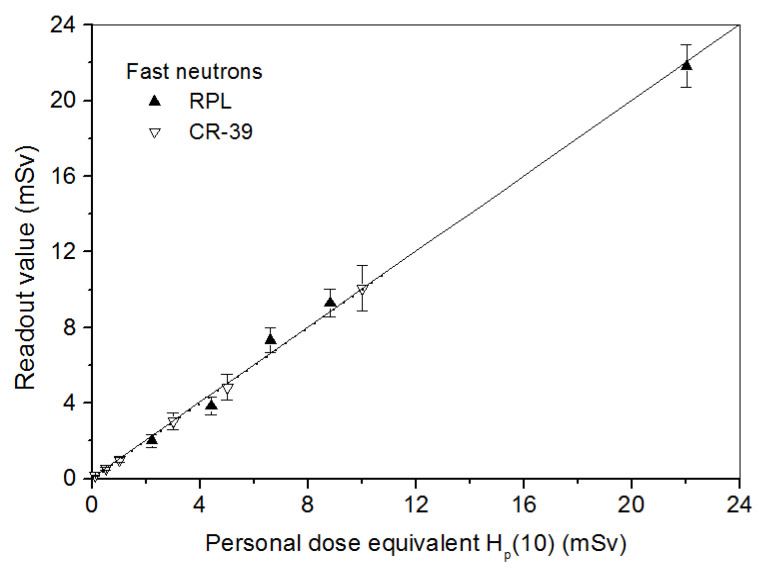
Linear responses of RPL and CR-39 detectors in terms of personal dose equivalent to fast neutrons at normal incidence.

**Figure 5 polymers-14-01801-f005:**
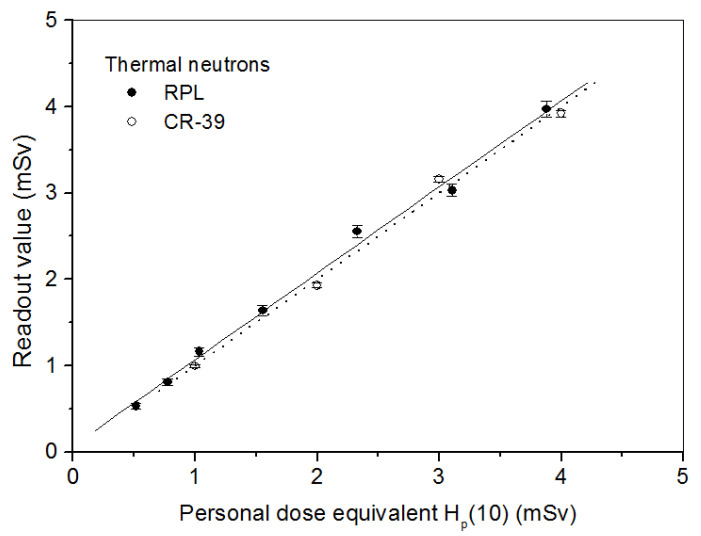
RPL and CR-39 detector responses at normal incidence to thermal neutrons as a function of personal dose equivalent H_p_(10).

**Figure 6 polymers-14-01801-f006:**
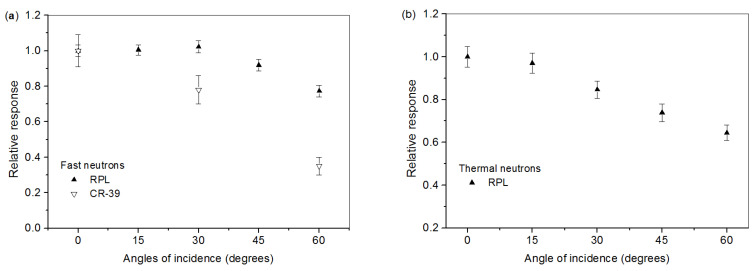
Angular variations of the RPL and CR-39 detectors: (**a**) fast neutrons; (**b**) thermal neutrons.

**Figure 7 polymers-14-01801-f007:**
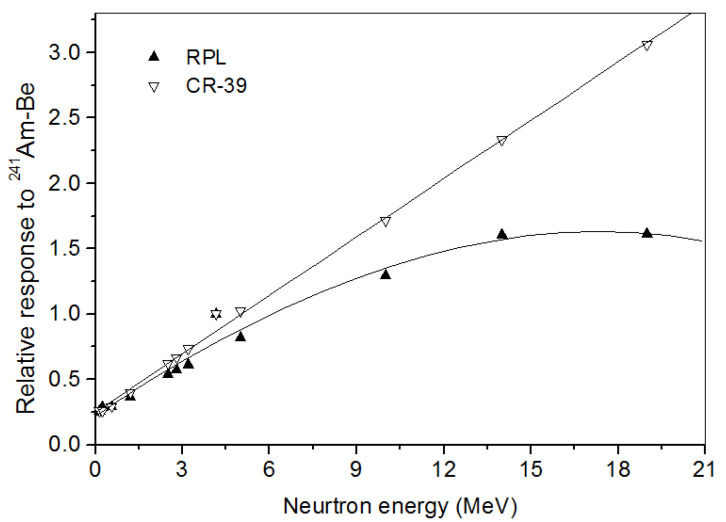
Relative energy response of the RPL and CR-39 detectors to fast neutrons.

**Table 1 polymers-14-01801-t001:** Reproducibility of measured values of RPL and CR-39 detectors for fast neutrons.

	RPL			CR-39		
H_p_(10) (mSv)	2.2	4.4	8.8	1	5	10
CV (%)	9.9	8.5	10.4	2.2	2.5	1.5

## Data Availability

Not applicable.

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
