# Peer review of "RPL Neutron Dosimetry in n-γ Fields in Comparison with Polymer Detectors Type CR-39"

_polymers, 2022, doi:10.3390/polym14091801_

Round 1
Reviewer 1 Report
The present paper reports the comparison results of neutron dosimetry with RPL glass detector and CR-39 plastic nuclear track detector. The subject is important for more feasible neutron dosimetry with RPL glasses, which may be replaced with CR-39. On the other side, this subject seems slightly out of scope of this journal, which covers “polymer” study. I believe the academic editor will make a decision whether this subject meets the scope. From my side, I provide some comments for improving the quality of this paper. The major issue is that several detailed explanations are missing, which should be clarified as a scientific research paper. Otherwise, it must be just “technical note”.
1) Line 54: “They Solid-State Nuclear Tracks Detectors (SSNTD) record their passage when their properties are compatible with detection energy thresholds.” requires something reference. I recommend: R.L. Fleischer, P.B. Price, R.M. Walker, Nuclear Tracks in Solids, University of California Press, Berkeley, USA, 1975.
2) Lines 62-65 and Fig.1 (right): Basically, your explanation is correct but Fig.1 (right) misleads somewhat because many etch pits in the microscope image seem to be over etched (large etch pits having bright spot in the center). In this case, the track registration sensitivity is not applicable anymore. It doesn’t matter for neutron dosimetry, of course, but more previse explanation should be given here. My suggestion is redrawing Fig.1 (midde), adding over etched case. Fig.1 in Tawara et al., Jpn. J. Appl. Phys., 47 (2008) 1726 (http://dx.doi.org/10.1143/JJAP.47.1726) may be helpful. The current Fig.1 (middle) (also left one) is a copy paste and I strongly recommend to redraw it anyway.
3) “4. Results and discussion”: Experimental condition is missing. How many detector pieces did you tested (number of samples)? How much dose did you irradiate? In Table 1, are H_p(10) values “irradiate dose” or “measured dose”?
4) Fig. 3: The unit of Readout value (vertical axis) is something dose [mSv]. Nobody understands how you obtained the dose results from RPL intensity in the glass and number of tracks in the CR-39. The detailed procedure to obtain the dose from each specific physics quantity should be given.
5) Fig. 3: Why is the error bar to be large as increase of dose? I suppose the statistical error should decrease with increase of given dose because RPL intensity and number of tracks increase.
6) Line 243: “RPL detectors were irradiated in the dose range from 2.2 to 22 mSv of IPHC 241Am-Be neutron source and the CR-39 in the dose range from 0.1 to 10 mSv”. Why were the dose ranges different for RRL glass and CR-39?
7) Line 316: “However, a critical angular dependence effect for Nuclear Track Detectors is known.” requires something references. I recommend: Somogyi & Szalay, Nucl. Instr. Meth. 109 (1973) 211 for the principle and Kodaira et al., Nucl. Instr. Meth. B383 (2016) 129 for the S dependency.
8) Fig. 6: “On the other hand, the RPL response shows a slight dependence on neutron energy beyond a 4.16 MeV.”. Why dose RPL response depend on the angular? More detailed explanation should be given.
9) Fig. 6 (b): CR-39 data is missing.
10) Line 342 and Fig. 7: Why dose RPL response depend on the energy? More detailed explanation should be given.
11) Fig. 7: How did you irradiate to neutrons with various energies?
Author Response
Dear collegue,
We would like to thank you for reviewing our manuscript.
Please find attached the responses to your comments and the revised version.
I remain at your disposal for any further information.
Best regards
A. Nourreddine

Reviewer 2 Report
The contribution describes new, interesting results on the RPL neutron dosimetry in n-gamma fields in comparison with CR-39 detectors. I have no important recommendations to modify and/or change the text of presentation. However, originally it was announced, thus I do accepted the paper
Author Response
Dear colleague,
We would like to thank you for reviewing our manuscript.
Please find attached the the revised version taking into account the remarks of the first reviewer.
Best regards
A. Nourreddine

Round 2
Reviewer 1 Report
Thanks for the revision. I have further minor comments:
1) Figures 1 and 2 come from other publication. I am not sure that you have already got copyright license for use of original figures but at least you need to give references in the captions as well.
2) I am not sure why you did not cite following paper about RPL angular dependency. I recommend to add it at the suitable location.
A. Nourreddine et al., "Study of a new neutron dosimeter incorporating RPL detectors", Radiat. Meas., 83 (2015) 47
3) My previous comment (Point 8): "Why dose RPL response depend on the angular? More detailed explanation should be given." is not addressed. This is not mandatory for dose assessment but I want to know the mechanism scientifically. I know the RPL efficiency is almost constant for angular irradiation of photons and heavy charged particles (but may be unpublished data...) and it is curious for neutrons specifically. I appreciate if you could give something comment about this.
Author Response
Dear colleague,
Thank you for your pertinent remarks, below are our answers.
Point 1 :
1) Figures 1 and 2 come from other publication. I am not sure that you have already got copyright license for use of original figures but at least you need to give references in the captions as well.
Response 1: the references has been added to the caption of figure 1.
Point 2:
I am not sure why you did not cite following paper about RPL angular dependency. I recommend to add it at the suitable location. A. Nourreddine et al., "Study of a new neutron dosimeter incorporating RPL detectors", Radiat. Meas., 83 (2015) 47
Response 2: This reference was introduced at the beginning of § 3.3 and added to the bibliographic references with number |44].
Point 3:
My previous comment (Point 8): "Why dose RPL response depend on the angular? More detailed explanation should be given." is not addressed. This is not mandatory for dose assessment but I want to know the mechanism scientifically. I know the RPL efficiency is almost constant for angular irradiation of photons and heavy charged particles (but may be unpublished data...) and it is curious for neutrons specifically. I appreciate if you could give something comment about this.
Response 3: We have added the following explanations at the end of § 4.
For more details, we recall that neutrons are indirect ionizing radiation and their detection is done via converters. In our case, fast neutrons were detected via (n,p) reactions in a polyethylene converter and thermal neutrons by (n,g) reaction in a Cd converter. Photons (indirect ionizing radiation) and protons (direct ionizing radiation) producing the directional dependencies of RPL detector are known in the literature. The RPL emitted radio-photoluminescent intensity, due to the activated luminescence centers in the glass during irradiation, is proportional to the amount of radiation received. The probability of having interactions over the entire depth of the RPL detector becomes significant for a normal incident. Indeed, the luminescence centers are created only in a surface layer of the exposed face of the RPL glass or over its entire depth. The absence of the dopant in silver in the first µm of the front face of the RPL detector will have for consequence a decrease of the RPL intensity when the angle of incidence increases. Also, the results for angular dependence correspond to a context of regulatory monitoring. For individual dosimetry, the dosimeter is to be worn on the human body. The calibration procedure must take this into account. So the dosimeter was irradiated on ISO water-filled phantom (ICRU, 1992, report 47) in terms of per-sonal dose equivalent. Results were obtained by rotating the dosimeter-phantom ensemble around a vertical axis perpendicular to the direction of the incident radiation.
Please find attached the latest version with the modifications in red.
I remain at your disposal for any further information.
Best regards
A. Nourreddine
